# The BREAK study protocol: Effects of intermittent energy restriction on adaptive thermogenesis during weight loss and its maintenance

**Filipa M. Cortez** [1] *, **Catarina L. Nunes**[2], **Luís B. Sardinha**[2], **Analiza M. Silva**[2], **Vítor H. Teixeira**[1,3,4,5]

**1** Faculty of Nutrition and Food Sciences, University of Porto, Porto, Portugal, **2** Exercise and Health Laboratory, CIPER, Faculty of Human Kinetics, University of Lisbon, Cruz-Quebrada, Portugal, **3** Research Centre of Physical Activity, Health and Leisure, CIAFEL, Faculty of Sport Sciences, University of Porto, Porto, Portugal, **4** Laboratory for Integrative and Translational Research in Population Health, ITR, Porto, Portugal, **5** Futebol Clube do Porto, Porto, Portugal

\* up201908421@edu.fcna.up.pt

**Data Availability Statement:** No datasets were generated or analysed during the current study. All relevant data from this study will be made available upon study completion.

## Abstract

### Background

Adaptive thermogenesis, defined as the decrease in the energy expenditure components beyond what can be predicted by changes in body mass stores, has been studied as a possible barrier to weight loss and weight maintenance. Intermittent energy restriction (IER), using energy balance refeeds, has been pointed out as a viable strategy to reduce adaptive thermogenesis and improve weight loss efficiency (greater weight loss per unit of energy deficit), as an alternative to a continuous energy restriction (CER). Following a randomized clinical trial design, the BREAK Study aims to compare the effects of IER versus CER on body composition and in adaptive thermogenesis, and understand whether participants will successfully maintain their weight loss after 12 months.

### Methods

Seventy-four women with obesity and inactive (20–45 y) will be randomized to 16 weeks of CER or IER (8x2 weeks of energy restriction interspersed with 7x1 week in energy balance). Both groups will start with 2 weeks in energy balance before energy restriction, followed by 16 weeks in energy restriction, then 8 weeks in energy balance and finally a 12-month weight maintenance phase. Primary outcomes are changes in fat-mass and adaptive thermogenesis after weight loss and weight maintenance. Secondary outcomes include weight loss, fat-free mass preservation, alterations in energy expenditure components, and changes in hormones (thyroid function, insulin, leptin, and cortisol).

**Funding:** Funding: F.M.C. was supported by Farmodiética S.A. https://www.farmodietica.com/en/home/ C.L.N. was supported with a PhD scholarship from the Portuguese Foundation for Science and Technology (SFRH/BD/143725/2019). https://www.fct.pt/en/ V.H.T. was supported by the Foundation for Science and Technology, through the FCT/UIDB/00617/2020 (CIAFEL) and LA/P/0064/2020 (ITR) Projects. https://www.fct.pt/en/ The funders did not and will not have a role in study design, data collection and analysis, decision to publish, or preparation of the manuscript.

## Discussion

We anticipate that The BREAK Study will allow us to better understand adaptive thermogenesis during weight loss and weight maintenance, in women with obesity. These findings will enable evidence-based decisions for obesity treatment.

## Trial registration

ClinicalTrials.gov: NCT05184361.

## 1. Introduction

Despite extensive research into lifestyle interventions for weight loss (WL) [1], one of the major challenges for treating obesity is WL maintenance [2], since weight regain rates are high [3]. Lack of long-term adherence to dietary and/or exercise recommendations has been pointed out as the major reason [4], however the existence of compensatory mechanisms in response to a prolonged negative energy balance (EB) may also play a role [5].

A decrease in resting energy expenditure (REE) is expected during a WL intervention due to fat-mass (FM) and fat-free mass (FFM) losses. However, some authors [6–8] showed that these reductions tend to be greater than predicted by WL, a phenomenon called "adaptive thermogenesis" (AT). Then, AT can function as a barrier to WL and contribute to weight regain [9, 10]. Still, some studies reported contrasting findings, as they did not find a significant value for AT even after a considerable WL [11–14]. This can be explained by a large heterogeneity in the methods used to quantify AT, a high variability in study designs, between subjects, and the wide magnitude of WL [15].

Furthermore, its relevance on long-term weight management (WM) has been recently questioned [16], as AT seems to be attenuated or even non-existent [15] after periods of (two to five weeks [16–19]) of weight stabilization.

Continuous energy restriction (CER), meaning a constant daily energy restriction (ER) according to individual's energy requirements, is the most common nutritional strategy for WL [20, 21]. However, some concerns have been pointed out to this strategy, since the behavioral, metabolic and endocrine adaptive responses it causes can compromise therapeutic adherence, undermining WL and WM [21–23]. These adaptive responses include an increased drive to eat, a reduced physical activity (PA) or energy cost of PA, a reduced energy expenditure (EE), and hormonal effects that facilitate the accumulation of adipose tissue and loss of lean tissues [24]. ER is associated with a reduction in thyroid hormones secretion (T3 and T4), variations in appetite-regulating hormones (decreased leptin, peptide YY, and increased ghrelin), variations in steroid hormones (increased cortisol), reduced insulin and testosterone, that all together influence EE, satiety and body composition [25, 26]. Simultaneously, some behavioral adaptations as a response to negative EB may occur, such as increasing sedentary behavior and/or decreasing light PA, more consistently observed in diet-only induced WL [26, 27].

Intermittent energy restriction (IER), which consists in interspersing periods of ER with periods of EB (the "refeed" or "diet breaks"), has been suggested as an alternative to CER [22]. The inclusion of periods of neutral EB is expected to attenuate adaptive responses to ER and WL in some regulatory hormones, which play a role in WL, satiety and REE [22, 24].

According to literature, there are 8 randomized clinical trials (RCT) comparing CER with IER in people with overweight/obesity, that achieved EB in the refeeding phases [6, 22, 23, 28–32]. Of these, only three resulted in a greater WL with IER [6, 23, 30]. Variability in study

design in these 8 RCT may have contributed to the different findings, favoring IER or not in comparison with CER. The study design differed in the duration of the intervention, the pattern of intermittency (days of ER vs. EB), the severity of calorie restriction, the food provision, the exercise recommendations, the adherence to the intervention, the sample size, and the baseline characteristics of the study population (sex and BMI).

Campbell et al. [23] investigated body composition changes in lean resistance trained individuals following a ~25% ER for 7 weeks in conjunction with 4-days/week resistance training, randomized to either IER or CER. IER cycled 5 days of ER with 2 days of EB using carbohydrate refeeds. The authors found that a 2-day carbohydrate refeed preserved FFM, dry FFM and REE during ER, compared to CER in resistance trained individuals. Davoodi et al. [30] compared IER and CER in 74 women with overweight and obesity for a six-week period. IER consisted of 3 cycles of 2 weeks (6-week total), and each cycle included 11 days of ER followed by 3 days of EB. IER was associated with a greater improvement in anthropometric measures, showed a better adherence to the dietary plan and a higher REE by the end of ER period, compared to CER. Byrne at al. [6] examined whether IER improved WL efficiency compared to CER in 51 men with obesity, for a 16-week ER period. IER group cycled 14 days of ER followed by 14 days of EB (30 weeks in total), with an ER of 33% in both IER and CER groups. The authors found a greater WL and fat loss with IER, and reported AT only for the CER group (~209 kJ/d) after a weight loss of ~8,4% of the initial weight, despite a greater weight loss in the IER group ~12,9%. As only men were included in the Byrne et al. study [6], there is a need to study the effects of a similar intervention in women. Furthermore, considering that the compensatory metabolic responses following ER and WL need a period of 7-to-14-days post-WL to be reversed [6, 25], and in order to maximize the ER/EB days´ ratio, another IER pattern should be considered. To our knowledge, this will be the first RCT comparing CER with IER (cycling 14 days of ER followed by 7-day EB periods), in women with obesity.

Pending this approach as a potential opportunity for obesity's treatment, this paper describes the protocol for a RCT, targeting to evaluate the effects of an IER, interspersing 14 days of ER with 7 days of EB, comparing to a CER. The primary aim of the trial is to assess whether IER will determine a greater FM loss and a reduced AT during WL and after 12-month maintenance, avoiding weight regain, compared to CER, in women with obesity and inactive. Secondary outcomes include WL, retention of FFM, alterations in EE components, and AT plasma-derived indices (thyroid function, insulin, leptin and cortisol).

## 2. Methods

### 2.1. Study design

The BREAK study is a RCT (allocation ratio of 1:1) that will be performed in adult women with obesity, randomly divided in 2 parallel groups: 1) CER and 2) IER. This study will include a three-phase intervention:

1. 2 weeks of neutral EB;

2. Active WL phase, where both groups will undergo 16 weeks of ER: CER—16 weeks of continuous ER; IER—2 weeks of ER interspersed with 1 week in EB, leading to a total of 23 weeks. IER length is 7-week longer than CER due to the 7x1-week of neutral EB, to maintain the same magnitude of ER in both interventions;

3. 8 weeks in neutral EB.

Participants will also be evaluated 12 months after the 3rd phase, to determine WL maintenance success. SPIRIT diagram is presented in Fig 1.

| | STUDY PERIOD | | | | | | | | |
| --- | --- | --- | --- | --- | --- | --- | --- | --- | --- |
| | Enrolment | Baseline | Post-allocation | | | | | | Close-out |
| TIMEPOINT* | $-t_1$ | $t_0$ | $t_2$ | $t_6$ | $t_{10}$ | $t_{14}$ | $t_{18}$ | $t_{26}$ | $t_{78}$ |
| **ENROLMENT** | | | | | | | | | |
| Eligibility screen | X | | | | | | | | |
| Informed consent | X | X | | | | | | | |
| Allocation | | X | | | | | | | |
| **NUTRITION INTERVENTIONS** | | | | | | | | | |
| Intermittent energy restriction | | ●————————————————————● | | | | | | | |
| Continuous energy restriction | | ●————————————————————● | | | | | | | |
| **ASSESSMENTS** | | | | | | | | | |
| Clinical data | X | | | | | | | | |
| Anthropometry (height*, body weight**) | | X*,** | X** | X** | X** | X** | X** | X** | X** |
| Body composition (DXA scan) | | X | X | | | | X | X | X |
| (Bioimpedance analysis) | | X | X | X | X | X | X | X | X |
| Resting energy expenditure (Indirect calorimetry) | | X | X | X | X | X | X | X | X |
| Physical activity (Accelerometry) | | X | X | X | X | X | X | X | X |
| Calculation of energy requirements | | X | X | X | X | X | X | X | X |
| Plasma hormones | | | X | | | | X | X | X |
| Contact with research team | | ●—————————————————————————————————● | | | | | | | |

**Fig 1. SPIRIT diagram.** *Time-points of the protocol: $-t_1$, enrolment; $t_0$, baseline; $t_2$, 2 weeks after baseline, start of energy restriction; $t_6$, 4 weeks in energy restriction; $t_{10}$, 8 weeks in energy restriction; $t_{14}$, 12 weeks in energy restriction; $t_{18}$, 16 weeks in energy restriction; $t_{26}$, 8 weeks in neutral energy balance; $t_{78}$, 52 weeks in weight maintenance.

The study was approved by the Ethics Committee of the Faculty of Nutrition and Food Sciences, University of Porto (Approval Number 31/2021/CEFCNAUP/2021) and will be conducted in accordance to the declaration of Helsinki for human studies from the World Medical Association [33]. It has been registered at www.clinicaltrials.gov (NCT05184361) prior to participants' recruitment.

All participants will be informed about the possible risks of this investigation before giving informed consent for enrollment in the study. Participants´ privacy and confidentiality will be ensured, during and after investigation, in accordance with the legislation in force.

## 2.2. Sample recruitment and selection

A total of 74 women with the identified criteria (Table 1) will be selected.

The disclosure of the clinical trial occurred after its registration in https://clinicaltrials.gov/, and was publicized in the media and social networks. Evaluations and consultations will be taken place at the Exercise and Health Laboratory, CIPER, Faculty of Human Kinetics, Lisbon, Portugal.

## 2.3. Screening process

Screening process will be phased, to identify and recruit eligible participants and give all the necessary information for an informed consent. During this process, procedures will be detailed, motivations accessed, and expectations anticipated (Table 2).

**Table 1. Eligibility criteria.**

**Inclusion criteria**

- Female sex;
- BMI 30–39.9 kg/m$^2$;
- Age 20–45 years;
- Weight stable (less than 5% of weight variation in the past 6 months);
- Inactive (less then 150 min/week of moderate PA or 75 min/week of vigorous PA [34]);
- Living in Lisbon metropolitan area;
- Available to be randomized to any of the trial groups;
- Willing to commit with the assigned group protocol, and being available for the 8 evaluations.

**Exclusion criteria**

- Previous or present major health disorders: cancer, autoimmune, chronic intestinal inflammatory, cardiac, psychiatric, kidney, and liver diseases (except liver steatosis), diabetes, or other medical conditions affecting energy balance;
- Menopause;
- Hormonal/thyroid disorders;
- Schizophrenia, bipolar disorder, or other psychotic disorders;
- Eating disorders;
- Major depression;
- Medications promoting weight gain or altering EB, including corticosteroids, antidepressants, anxiolytics, mood stabilizing, and antipsychotics;
- Medicines/dietary supplements for weight-loss in the previous 3 months;
- History of bariatric surgery/liposuction procedures;
- Pregnant for the previous 6 months/breastfeeding;
- Planning to get or getting pregnant in the next 2 years;
- Surgery/hospital admission in the previous month.
- Current consumption of more than 14 alcoholic drinks per week or abuse of other substances, and/or current acute treatment/rehabilitation program for alcohol/substance abuse.

**Table 2. Screening process overview.**

| Screening step 1—email and/or phone evaluation |
| --- |
| • Provide general information about the study; |
| • Screen eligibility criteria. |

| Screening step 2—online one-to-one interview |
| --- |
| • Validation of the eligibility criteria and all the information provided in screening step 1; |
| • Assess weight loss motivation; |
| • Assess lifestyle logistics (home, work, leisure, planned holidays); |
| • Provide detailed information about the study goals, procedures, timing of evaluations, length of the intervention and weight management phase; |
| • Confirm availability for being present at the clinical trial 8 evaluation moments; |
| • Assess food preferences, food intolerances and allergies, as well as food habits using a 24h-dietary recall on weekdays and weekend; |
| • Discuss the nutritional intervention and recommendations; |
| • Ensure availability for being randomized to any of the two groups; |
| • Clarify any questions participants may have; |
| • Provide one copy of the informed consent; |
| • Schedule the 1st visit/evaluation of the study. |

## 2.4. Randomization

Participants will be randomized to one of the two arms of the study through a simple automatic randomization scheme generated by computer, controlled by the researcher responsible for the data treatment (which is not the main investigator). Randomization will be performed before the 1st visit/evaluation. Participants will be informed of their assigned group during the 1st visit/evaluation.

## 2.5. Calculation of energy stores

Considering the principle of energy conservation [5, 35, 36], the rate of change in body energy storage (ES) is equal to the difference between the rates of energy intake (EI) and EE, expressed as energy per unit of time [35].

EB equation is the following:

$$ES\ (kcal.d^{-1}) = EI - EE.$$

When the EI surpasses the EE, changes in ES will be positive, leading to a positive EB. On the other hand, a negative EB will be created when the EI is lower than the EE.

EB can be calculated from the change in body energy stores from the beginning to the end of the WL intervention. Therefore, using the established energy densities of 1.0 kcal.g$^{-1}$ for FFM and 9.5 kcal.g$^{-1}$ for FM, the following equation will be used to quantify the average rate of changed body energy stored or lost in kilocalories per day:

$$EB(kcal/d) = 1.0\frac{\Delta FFM}{\Delta t} + 9.5\frac{\Delta FM}{\Delta t},$$

Where $\Delta$FFM and $\Delta$FM represent the change in grams of FFM and FM, respectively, from the beginning to end of the intervention and $\Delta t$ is the time length of the intervention in days [37–39].

## 2.6. Nutritional intervention and estimation of energy requirements

Nutritional intervention will comprise a personalized dietary plan prescribed for each participant, considering their daily energy requirements (DER) through each phase of the intervention. The DER will be estimated by multiplying the REE measured (mREE) by a PA level (PAL) [40], in phase 1, and using REE and accelerometry data [41], in phase 2 and 3. DER will be calculated in every visit/evaluation performed during the trial (8 in total). The main investigator will be responsible for all appointments, follow-ups and diet plans, including calculation of DER.

The energy distribution by macronutrients will be 35% from protein, 35% from carbohydrate and 30% from fat in all study-phases. A Mediterranean-style diet will be prescribed for both groups, aiming for improving diet quality, and using portion control to achieve energy requirements during EB, ER and WM. The Mediterranean-style diet includes the following recommendations: high intake of vegetables including leafy green vegetables, fruits, wholegrain cereals, nuts and pulses, legumes, and extra virgin (cold pressed) olive oil; moderate intake of fish, seafood, eggs, poultry, and dairy products; low intake of red meat (less than twice a week) and red wine should be consumed in moderation [42]. Processed foods, sweets, cookies, chips, high-fat cheeses, sausages and unhealthy foods are to be avoided. The dietary plan will be adjusted when necessary, according to participant´s feedback.

**2.6.1. Neutral energy balance for weight stabilization.** For weight stabilization, a personalized dietary plan meeting 100% of their DER will be prescribed for each participant. This dietary plan will take into consideration calculations of DER at every visit/evaluation, and will be adjusted when needed.

Participants will be provided with a digital scale on the 1st visit, to track their weight daily at home [43, 44], and will be instructed to weight themselves after a 10h-overnight fast, wearing only underwear, and weekly share with the main investigator their overnight fasting body weight. During EB, if they identify a weight gain of more than 1 kg, participants will have clear instructions on how to adjust their dietary plan, in order to reduce and stabilize weight [6].

**2.6.2. Energy restriction for weight loss.** For both groups, an ER of 33% of one's DER will be created [6, 32] to achieve WL. Participants will have their personalized dietary plan adjusted every 4 weeks during the ER phase, according to REE measurements and PA monitoring, to assure the same energy deficit throughout this phase.

During this phase, participants will continue to track their weight daily at home [43, 44], and weekly share with the main investigator their overnight fasting body weight.

**2.6.3. Energy intake and diet compliance.** Compliance to the dietary plan will be monitored once a week by phone, and participants will be asked to send, via social media platforms, daily photo food records during the intervention [45].

**2.6.4. Adherence to diet.** Participants from both groups will undergo an ER meeting 33% of their DER.

Adherence will be assessed through the following equation proposed by Racette et al [46]:

$$Adherence_{(\%)} = 100 \times \left[ \left( 1 - \frac{EI_{4mo}}{EI_{baseline}} \right) \times \frac{100}{ER_{prescribed(\%)}} \right]$$

Where the EI will be calculated by the "intake-balance method" [47], through changes in FM and FFM, together with total daily EE (TDEE). The following equation was used:

$$EI_{(kcal/d)} = EE_{(kcal/d)} + EB_{(kcal/d)},$$

Where EE represents the total daily EE measured by accelerometry, and EB calculated through

changes in FM and FFM. The degree of ER during the WL phase will also be calculated through this equation. It will be also assessed if participants from IER group not only accomplished the prescribed ER but also if underwent the 1-week periods of neutral EB.

## 2.7. Physical activity

To enroll this study, as stated in inclusion criteria, participants needed to be considered inactive (less then 150 min/week of moderate PA or 75 min/week of vigorous PA [34]). This PA level should be maintained throughout the WL and WM phase.

## 2.8. Retention of the participants

All efforts will be made before, during and after the intervention to retain participants in the study, in order to manage costs, equipment, human resources and data colleting time.

**2.8.1. Strategies to engage participants, avoid low attendance and dropouts.**   During the intervention, and throughout the 8 visits to the laboratory for assessment and consultations, participants will be reminded of the importance of adhering to the recommendations, as well as attending appointments. Some flexibility will be taken into consideration, as long as it does not interfere with the study plan/goals. A 1-week adjustment will be considered as an on-time assessment, maintaining the previously defined length of each study phase.

Between the scheduled visits and consultations, additional contacts will be held by phone, following up results and adherence to the dietary plan.

After the intervention phase is over (7th visit), and the 12-month WM phase starts, the research team will remain available to follow-up the participants, with at least one phone/video consultation per month.

**2.8.2. Rewards for participation.**   Participants will not receive any type of financial incentive. Nevertheless, they will benefit from a free nutritional intervention for WL and WM, which includes 8 nutrition appointments and weekly remote follow-up by a certified clinical dietitian for 78 weeks (CER group), or 85 weeks (IER group).

## 3. Measurements

Body weight, body composition and resting energy expenditure (REE) will be collected at 8 different moments, and DXA scans at 5 different moments, as described in Fig 2. Participants will be informed not to take diuretics in the previous 7 days, have alcoholic drinks/coffee in the previous 24 hours, and perform vigorous PA 24-hours prior to the laboratory visits, and to urinate 30 minutes before evaluations [48, 49]. All evaluations will be performed with a minimum 10-hour overnight fast, barefoot and wearing underwear and a disposable vest, to preserve participants comfort.

### 3.1 Body composition

**3.1.1. Anthropometry.**   Anthropometry will consider the procedures and recommendations described in ISAK guidelines [50]. Weight and height will be determined using a digital scale with a stadiometer Seca 704s, with 0,1 kg and 0,1 cm intervals (Seca, Hamburg, Germany). Body mass index (BMI) will be calculated using the formula $[weight (kg)/height^2(m^2)]$ and the cut-off points of the World Health Organization (WHO) will be used [51].

**3.1.2. Dual-Energy X-ray Absorptiometry.**   In order to assess body composition stores–Fat Mass (FM) and Fat-Free Mass (FFM), a whole-body dual energy X-ray absorptiometry (DXA) scan (Hologic Explorer-W, Waltham, USA) will be used. All the assessments will be performed by the same investigator. Total abdominal fat, which includes intra-abdominal fat

## Intermittent energy restriction

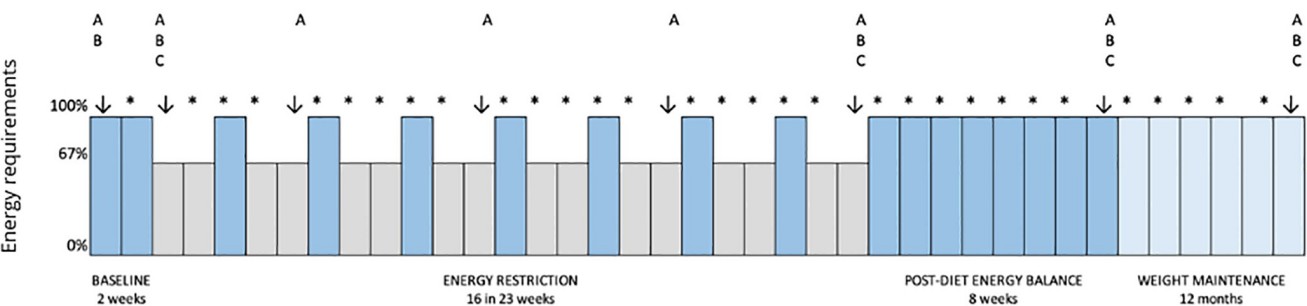

## Continuous energy restriction

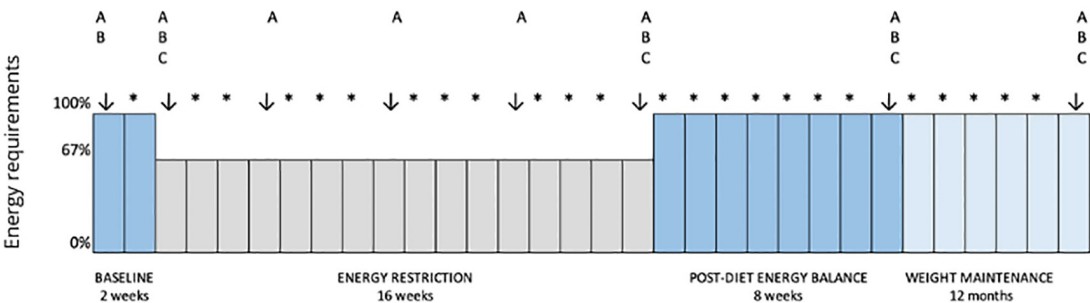

**Fig 2. Schematic description of the study design.** Arrows indicate points for measurements. A—body weight, body composition using bioimpedance analysis (BIA), resting energy expenditure (REE), physical activity (PA) using accelerometry, and calculation of daily energy requirements (DER). B—body composition using Dual-Energy X-ray Absorptiometry (DXA). C—determination of plasma hormones (cortisol, insulin, leptin, thyroid free T3 and T4). * —phone/video contact with research team.

plus subcutaneous fat, will be distinguished by identifying a specific region of interest (ROI) within the analysis program. Specific DXA ROIs for abdominal regional fat will be defined as follows: from ROI 1, the upper edge of the second lumbar vertebra (approximately 10 cm above the L4 to L5) to above the iliac crest and laterally encompasses the entire breadth of the abdomen, thus determining total abdominal fat mass [52].

**3.1.3. Bioimpedance analysis (BIA).** Bioimpedance analysis will be performed using BIA-101 BIVA PRO (Akern srl, Florence, Italy). Before the test, subjects will be instructed to lie in a supine position with their arms and legs abducted at a 45 angle for 10 min [53]. Four electrodes will be placed on the dorsal surfaces of the right foot and ankle, as well as the right wrist and hand.

This equipment will measure the resistance (Rz) and reactance (Xc) using a 250 mA alternating current at 50 kHz ± 1%. Calibration will happen every morning according to be manufacturer instructions. Bioelectrical phase angle will be calculated as the tangent arc of Xc/R x 180º/ π. Vectorial analysis of bioimpedance will use BIVA method, normalizing R and Xc for height in meters. FM and FFM percentage will be determined using Bodygram® software (AkernSrl., Florence, Italy) [54].

### 3.2. Resting energy expenditure

REE will be determined using indirect calorimetry COSMED Fitmate (Cosmed, Rome, Italy) using a face mask. Fitmate is a metabolic analyzer designed for measurement of oxygen

consumption and EE during rest and exercise. It uses a turbine flowmeter for measuring ventilation and a galvanic fuel cell oxygen sensor for analyzing the fraction of oxygen in expired gases. REE is calculated from oxygen consumption, at a fixed respiratory quotient of 0.85, and estimated grams of urinary nitrogen using a modified Weir equation [55, 56].

The metabolic analyzer will be calibrated in the morning before testing, using a pumping gas with 3 L calibration syringe through the flow meter, according to the manufacturer's recommendations. The test will be performed early morning (8 a.m. to 11 a.m.), after a minimum 10h-overnight fast. Individuals will be advised to reduce PA the most until the test. All measurements will happen in a thermic neutral environment (approximately 22°C and humidity between 40–50%) [53], a quiet and dimly lit room, in semi-recumbent position [55]. During the test, participants will be asked to relax and keep immobile, without doing any activities, such as fidgeting, reading, listening to music, talking, nor falling asleep [57]. Rest duration will be 15 minutes, followed by a 30-minute test duration, ignoring the first 10 minutes [58]. Fitmate steady state is achieved when the CV in VO2 is <10% during the 30-minute measurement.

**3.2.1. Adaptive thermogenesis.** The REE will be predicted (pREE) through linear regression analysis, with the baseline measured REE as a dependent variable, and FM (kg) and FFM (kg) as independent variables. The predictive equation will be used to assess pREE at each time point, using the FM and FFM values of the respective time.

Adaptive thermogenesis (AT) will be assessed by the following equation:

$$AT(kcal/d) = (mREE \text{ at the end of the intervention} - pREE \text{ at the end of the intervention})$$
$$- (mREE \text{ baseline} - pREE \text{ baseline}),$$

where negative values will indicate a lower-than-expected REE due to body composition changes [59].

## 3.3. Free-living physical activity and total energy expenditure

PA will be determined using ActiGraph wGT3X-BT accelerometer (ActiGraph, Pensacola FL, USA), which expresses minutes per day spent in different activities. Accelerometers will be placed on the right hip close to the iliac crest, and activated when participants go to the laboratory visit, being used during 7 days. The devices must be used while participants are awake, and will only be asked to be removed during water activities, such as shower and swimming. Participants will be asked to register the time and reason each time they take off the accelerometer. The activation of the devices, download and processing will be held with Actilife software (v.6.9.1).

The cutoff values used for defining PA intensity and the average time at each intensity level (sedentary or light, moderate, hard and very hard PA intensities) will be as follows: sedentary or light: <2689 counts per minute$^{-1}$ (<3.00 METs); moderate: 2690–6166 counts per minute$^{-1}$ (3.00–5.99 METs); hard: 6167–9642 counts per minute$^{-1}$ (6.00–8.99 METs); very hard: >9642 counts per minute$^{-1}$ (>8.99 METs) [60].

Among adults, at least 3–5 days of monitoring are required to estimate usual PA [61], therefore participants will be included if they show a minimum of three valid days of accelerometer data. A valid-day will be defined as having 600 or more minutes ($\geq 10$ h) of monitor wearing, corresponding to the minimum daily use of the accelerometer.

TDEE will be calculated as the sum of REE, thermic effect of food (TEF) and PA energy expenditure (PAEE) [62]. PAEE data will be collected through accelerometry using Crouter and colleagues' equations [61, 63, 64].

### 3.4. Plasma hormonal determination

Plasma cortisol, insulin, leptin and thyroid free-T3 and T4 will be determined for AT analysis in 4 moments (Fig 1). Measurements of plasma thyroid levels (free-T3 and T4) and cortisol will be determined by immunoassay with chemiluminescence detection (Advia Centaur, Siemens). Insulin assessment will be performed in an automated analyser with chemiluminescence detection (Advia Centaur, Siemens), and leptin plasma levels by enzyme immunoassay (ELISA). Reference values for these parameters will be considered.

### 3.5. Statistics

Statistical analysis will be performed using SPSS statistics software version 29.0, 2022 (SPSS Inc., an IBM Company, Chicago IL, USA), with a statistical significance set at $p < 0.05$ (2-tailed).

Descriptive statistics will be calculated (mean, standard deviation, and range). Linear mixed models will be used to assess outcomes for the impact of group, time and group*time interaction, including the randomized group (control vs intervention group) and time (baseline at the 2-week EB, at the start and every 4-week of ER intervention phase, at the start and at the end of the 8-week post-diet EB phase, and at the end of WM phase) as fixed effects. When necessary, adjustments for confounding variables (covariates) will be considered. The covariance matrix for repeated measures within subjects over time will be modelled as unstructured or, if necessary, compound symmetry. The normality of model residual distributions will be examined graphically and with the Kolmogorov-Smirnov test. Difference-in-differences between IER and CER throughout time will be assessed by performing contrasts [Difference-in-differences (DiD)], calculated as the difference between changes for IER (Difference_IER = T2 -T1) and changes for CER (Difference_CER = Time2 –Time1): DiD = (Difference_IER)–(Difference_CER).

All analysis will be intention-to-treat, including data from all the participants who will assign in this study. Sensitivity analyses will be carried out for some variables of interest, by using single imputation of missing data to predict missing outcomes from demographics and baseline measures.

**3.5.1. Power sample calculation.**   For sample and power calculations, this study is powered based on changes in total body fat assessed by DXA. Considering a type I error of 5% and a power of 95% to detect differences at the dependent variable, with a statistical significance and an effect size of 0.93 (differences in FM (kg) were considered following Byrne et al. results [6]), a total of 26 participants per group will be needed (using GPower software version 3.1.9.6). Considering a 30% drop-out rate [6] throughout the study, we will recruit 74 participants (37 in each group).

**3.5.2. Trial status.**   Recruitment for this clinical trial started on January 13th, 2022, and is expected to end on June 30th, 2024.

## 4. Discussion

This RCT aims primarily to evaluate the effects of an IER, interspersing 14 days of ER with 7 days of EB, on body composition (body weight, FM and FFM), and more specifically on AT, during WL and WM phase. It also aims to understand whether participants from both groups (IER and CER) will successfully maintain their WL 12 months after completion of the intervention.

Secondary objectives of this study are the following: (i) to compare the effects of IER and CER on WL, FM loss, and preservation of FFM and REE during the intervention phase; (ii) to determine which group is more successful in the 12-month WM phase and in improving body

composition profile (higher FM loss with best preservation of FFM); (iii) to analyze if AT is maintained during WM phase; (iv) to analyze the impact of AT in WM phase, regarding the hormonal adaptation (plasma hormones: cortisol, insulin, leptin, thyroid free T3 and T4).

A 5% WL has been pointed out to be enough to improve metabolic parameters (glycemic measures and triglycerides, blood pressure, HDL and LDL cholesterol) and is currently a standard goal in WL interventions [65]. Nevertheless, a greater reduction of 10% or more can lead to maximal health benefits, reducing many of the comorbidities associated with obesity [66, 67]. In fact, according to Wing and Hill [68], successful WL maintainers should be defined as "individuals who have intentionally lost at least 10% of their body weight and kept if off at least 1 year". With The BREAK Study we expect to find a clinically significant weight loss [69] of 10% in both groups, considering the length of the ER intervention, and the magnitude of ER [70].

According to Byrne and colleagues [6, 25, 71], adaptive responses to ER and WL can be reversed by a 7-to-14-day period of EB after, in adults with overweight or obesity. In "MATADOR" Study, a IER using a 2-week EB periods improved WL efficiency compared with CER, being AT only found in CER group, despite the lower WL and FM loss [6]. In review by Peos and colleagues [21], it was stated that "refeeds" with at least 7 days attenuate the adaptive responses to longer-term periods of ER, potentially by attenuating the reductions in EE. Furthermore, these authors also mentioned that adopting refeed periods in IER may provide a mental break from extended periods of ER, leading to a higher long-term adherence to the dietary plan compared to CER. Therefore, we also anticipate a major weight and FM loss in IER group, comparing to CER, with a greater retention of FFM, reducing therefore AT [6, 23, 30], due to the inclusion of 7-day breaks to restore EB every 2-weeks of ER, which can minimize the compensatory mechanisms associated with ER and WL, such as changes in appetite-related hormones [24] and some behavioral compensations such as increases in sedentary behavior [26].

During WL phase, we expect to find a reduction in all EE components, as usually seen during ER, namely REE and non-resting EE (spontaneous PA (SPA), non-exercise PA (NEPA), and exercise PA (EPA)) [72].

WL usually causes a reduction in thyroid hormones (T3 and T4), insulin and leptin, increasing appetite [7, 73], and causes an increase in cortisol, which reduces EE [22]. We expect to find a lower reduction in insulin, leptin, thyroid T3 and T4, as well as a lower increase in cortisol in IER, comparing to CER. The minimization of these compensatory adaptations can be explained by the 1-week EB every 2-weeks of ER. After ER intervention, we anticipate a successful WL maintenance in both groups, possibly greater in the IER group, as well as a lower AT.

The BREAK Study focuses not only in reducing energy intake through portion control, but also in improving nutritional quality of the diet, by increasing fruit and vegetable intake, and decreasing processed and energy-dense foods. Participants will be educated and encouraged to daily monitor their behavior, weight, and eating pattern, leading to self-efficacy for diet and WM, which are determinants of WL maintenance [66]. Although maintaining high levels of PA has been pointed out as a determinant of WL maintenance [74], the BREAK Study is a diet-only [75] intervention. Therefore, no PA recommendations will be given to participants throughout the WL and WM phases.

One of the strengths of The BREAK Study is the 2-week baseline EB for determining energy requirements, stabilizing weight, and adapting participants to the prescribed dietary plan. Another strength is the determination of REE and PA monitoring every 4 weeks of ER, allowing adjustment of the dietary plan to ensure the energy deficit of 33% during the 16-week ER

phase. Monitoring AT-related hormone changes during WL and WM is also a strong point of this study, enabling a better understand of AT in both groups.

However, some limitations should be addressed, such as: i) the nutritional intervention occurs in a free-living scenario, with no food being delivered to participants, preventing a valid assurance that participants are in fact consuming the prescribed energy. To minimize this adherence limitation, participants are instructed to weekly share with the main investigator their overnight fasting body weight and photo records of their meals; ii) a large interindividual variability in PAEE is expected and may not be accurately detected by accelerometry; iii) the eventual more than expected drop out of the study, due to its length, possibly weakening the statistical power of the analysis.

We anticipate that The BREAK Study will allow us to better understand AT during WL and WM interventions in women with obesity. Moreover, we expect to find a successful alternative to CER, enabling more tailored nutritional interventions, according to individuals needs and lifestyle. This study will also allow participants to lose weight and FM, while attenuating AT, improving their metabolic health, and encouraging them to adhere to a healthy lifestyle and acquire nutritional knowledge that will facilitate WM in the long-term. Finally, the findings of this trial will enable evidence-based decisions for the treatment of obesity.

## Supporting information

**S1 Checklist. SPIRIT checklist.**
(DOC)

**S1 File. Ethics Committee opinion—Original.**
(PDF)

**S2 File. Ethics Committee opinion—English.**
(PDF)

**S3 File. Ethics Committee project summary—Original.**
(PDF)

**S4 File. Ethics Committee project summary—English.**
(PDF)

**S1 Appendix. Informed consent for participants—Original.**
(PDF)

**S2 Appendix. Informed consent for participants—English.**
(PDF)

## Acknowledgments

The authors express their gratitude to all the participants involved in this study.

## Author Contributions

**Conceptualization:** Filipa M. Cortez, Vítor H. Teixeira.

**Data curation:** Filipa M. Cortez, Catarina L. Nunes.

**Funding acquisition:** Filipa M. Cortez.

**Investigation:** Filipa M. Cortez.

**Methodology:** Filipa M. Cortez, Analiza M. Silva, Vítor H. Teixeira.

**Project administration:** Filipa M. Cortez.

**Resources:** Filipa M. Cortez, Luís B. Sardinha, Analiza M. Silva.

**Software:** Filipa M. Cortez.

**Supervision:** Analiza M. Silva, Vítor H. Teixeira.

**Visualization:** Filipa M. Cortez.

**Writing – original draft:** Filipa M. Cortez, Catarina L. Nunes.

**Writing – review & editing:** Filipa M. Cortez, Catarina L. Nunes, Luís B. Sardinha, Analiza M. Silva, Vítor H. Teixeira.

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
