## [Decision Letter · Decision Letter 0]

17 Sep 2023

PONE-D-23-12945The BREAK Study Protocol: Effects of intermittent energy restriction on adaptive thermogenesis during weight loss and its maintenancePLOS ONE

Dear Dr. Cortez,

Thank you for submitting your manuscript to PLOS ONE. After careful consideration, we feel that it has merit but does not fully meet PLOS ONE’s publication criteria as it currently stands. Therefore, we invite you to submit a revised version of the manuscript that addresses the points raised during the review process.

We look forward to receiving your revised manuscript.

Kind regards,

Vitor Barreto Paravidino

Academic Editor

PLOS ONE

Journal Requirements:

2. Please amend your authorship list in your manuscript file to include authors Filipa M Cortez, Catarina L Nunes, Vítor H Teixeira, Analiza M Silva and Luís B Sardinha.

3. We note that Figure 2 in your submission contain copyrighted images. All PLOS content is published under the Creative Commons Attribution License (CC BY 4.0), which means that the manuscript, images, and Supporting Information files will be freely available online, and any third party is permitted to access, download, copy, distribute, and use these materials in any way, even commercially, with proper attribution. For more information, see our copyright guidelines: http://journals.plos.org/plosone/s/licenses-and-copyright.

4. We note that the original protocol that you have uploaded as a Supporting Information file contains an institutional logo. As this logo is likely copyrighted, we ask that you please remove it from this file and upload an updated version upon resubmission.

Reviewers' comments:

Reviewer's Responses to Questions

**Comments to the Author**

1. Does the manuscript provide a valid rationale for the proposed study, with clearly identified and justified research questions?

Reviewer #1: Yes

Reviewer #2: Yes

2. Is the protocol technically sound and planned in a manner that will lead to a meaningful outcome and allow testing the stated hypotheses?

Reviewer #1: Yes

Reviewer #2: Yes

3. Is the methodology feasible and described in sufficient detail to allow the work to be replicable?

Reviewer #1: Yes

Reviewer #2: Yes

4. Have the authors described where all data underlying the findings will be made available when the study is complete?

Reviewer #1: No

Reviewer #2: Yes

5. Is the manuscript presented in an intelligible fashion and written in standard English?

Reviewer #1: Yes

Reviewer #2: Yes

6. Review Comments to the Author

You may also provide optional suggestions and comments to authors that they might find helpful in planning their study.

Reviewer #1: I would like to congratulate the authors for the project. I take the opportunity and wish them success in completing the study.

Reviewer #2: The authors aim to assess whether intermittent energy restriction will determine greater fat mass loss and lower adaptive thermogenesis during weight loss and maintenance. The sample size, methods, and design are correct, as are the stats. The authors documented and discussed their hypothesis correctly.

Minor revision with the following remarks:

- The introduction mentions that there are 8 randomized clinical trials (RCTs) that compare CER with IER (lines 102-104), however, only the results from studies that have found greater weight loss with IER have been detailed. It would be helpful to know the outcomes of other studies and explore potential reasons for the conflicting results.

- The description of the REE measurement protocol, did not include information on rest time before measurement, on the calibration of the calorimeter, and the REE validity criterion (steady-state or %CV VO2).

- As described by Nunes et al. 2022, the AT equation must be REEm -REEp (subtracting REEp from REEm), to allow the interpretation that negative values mean lower-than-expected REE due to body composition changes. In lines 365-366, the equation is inverted.

- Cutoff values that define the AF intensity and the average time at each intensity level will be based on non-triaxial accelerometers (line 382)? Sasaki et al. 2011 (DOI: 10.1016/j.jsams.2011.04.003) proposed tri-axial vector magnitude (VM3) cut-points to classify physical activity (PA) intensity.

7. PLOS authors have the option to publish the peer review history of their article (what does this mean?). If published, this will include your full peer review and any attached files.

Reviewer #1: No

Reviewer #2: **Yes: **Tatiana Almeida de Moraes Campos

---

## [Author Response · Author response to Decision Letter 0]

25 Sep 2023

Dear Editor and Reviewers,

We would like to express our sincere appreciation for the time you devoted to this review process. Your insights and feedback have played an indispensable role in refining our work, and enhancing its quality. 

Considering the Journal Requirements

Reply: We have consulted the suggested files, and proceeded as required.

2. Please amend your authorship list in your manuscript file to include authors Filipa M Cortez, Catarina L Nunes, Vítor H Teixeira, Analiza M Silva and Luís B Sardinha.

Reply: We changed it accordingly.

3. We note that Figure 2 in your submission contain copyrighted images. All PLOS content is published under the Creative Commons Attribution License (CC BY 4.0), which means that the manuscript, images, and Supporting Information files will be freely available online, and any third party is permitted to access, download, copy, distribute, and use these materials in any way, even commercially, with proper attribution. 

Reply: Considering we were unable to request permission for the copyrighted images, we removed them from the submitted Figure, without affecting the understanding of its content.

4. We note that the original protocol that you have uploaded as a Supporting Information file contains an institutional logo. As this logo is likely copyrighted, we ask that you please remove it from this file and upload an updated version upon resubmission.

Reply: The logo was removed from the file, as required.

Reply: We reviewed the reference list, and no papers have been retracted.

Considering the Reviewers' comments

Have the authors described where all data underlying the findings will be made available when the study is complete?

Reviewer #1: No; Reviewer #2: Yes

Reply: We have updated Data Availability information in the manuscript, as required.

(page 2, lines 38-45)

Review Comments to the Author

Minor revision with the following remarks

1. The introduction mentions that there are 8 randomized clinical trials (RCTs) that compare CER with IER (lines 102-104), however, only the results from studies that have found greater weight loss with IER have been detailed. It would be helpful to know the outcomes of other studies and explore potential reasons for the conflicting results.

Reply: We appreciate the reviewer comment, and therefore included in the text the potential reasons for the conflicting results. We did not describe the outcomes and main characteristics for the remaining studies due to length of introduction.

(page 5-6, lines 121-126)

2. The description of the REE measurement protocol, did not include information on rest time before measurement, on the calibration of the calorimeter, and the REE validity criterion (steady-state or %CV VO2).

Reply: We have added the required information in the manuscript.

(page 17-18, lines 376-378, 384-387)

3. As described by Nunes et al. 2022, the AT equation must be REEm -REEp (subtracting REEp from REEm), to allow the interpretation that negative values mean lower-than-expected REE due to body composition changes. In lines 365-366, the equation is inverted.

Reply: We have changed it accordantly.

(page 18, lines 395-396)

4. Cutoff values that define the AF intensity and the average time at each intensity level will be based on non-triaxial accelerometers (line 382)? Sasaki et al. 2011 (DOI: 10.1016/j.jsams.2011.04.003) proposed tri-axial vector magnitude (VM3) cut-points to classify physical activity (PA) intensity.

Reply: We appreciate the reviewer comment and would like to explain our rationale. The accelerometer allows to select the counts of the vertical axis (axis 1) to classify the levels of intensity based on Troiano's cutoff values (1). By using this option, we would be able to compare our results with other studies that used Troiano's cutoffs. Regardless, we are also able to select the vector magnitude based on the 3 axis and classify the level of intensity based on Freedson cut-points. Considering the reviewer's point based on the paper of Sasaki et al. 2011, we changed the cutoff values for those proposed by these authors (2). 

(page 19, lines 416-419)

7. PLOS authors have the option to publish the peer review history of their article. If published, this will include your full peer review and any attached files.

Reply: We opt to publish peer-reviewed history of the article.

References

1. Troiano RP, Berrigan D, Dodd KW, Mâsse LC, Tilert T, McDowell M. Physical activity in the United States measured by accelerometer. Med Sci Sports Exerc. 2008;40(1):181-8.

2. Sasaki JE, John D, Freedson PS. Validation and comparison of ActiGraph activity monitors. J Sci Med Sport. 2011;14(5):411-6.

---

## [Editor Report · Decision Letter 1]

26 Oct 2023

The BREAK Study Protocol: Effects of intermittent energy restriction on adaptive thermogenesis during weight loss and its maintenance

PONE-D-23-12945R1

Dear Dr. Cortez,

We’re pleased to inform you that your manuscript has been judged scientifically suitable for publication and will be formally accepted for publication once it meets all outstanding technical requirements.

Kind regards,

Vitor Barreto Paravidino

Academic Editor

PLOS ONE
---

## [Editor Report · Acceptance letter]

3 Nov 2023

PONE-D-23-12945R1 

The BREAK Study Protocol: Effects of intermittent energy restriction on adaptive thermogenesis during weight loss and its maintenance 

Dear Dr. Cortez:

I'm pleased to inform you that your manuscript has been deemed suitable for publication in PLOS ONE. Congratulations! Your manuscript is now with our production department. 

Kind regards, 

on behalf of

Dr. Vitor Barreto Paravidino 

Academic Editor

PLOS ONE